# Focus on Mitochondrial Respiratory Chain: Potential Therapeutic Target for Chronic Renal Failure

**DOI:** 10.3390/ijms25020949

**Published:** 2024-01-12

**Authors:** Yi Wang, Jing Yang, Yu Zhang, Jianhua Zhou

**Affiliations:** Department of Pediatrics, Tongji Hospital, Tongji Medical College, Huazhong University of Science & Technology, Wuhan 430030, China; wangyidoctormed@163.com (Y.W.); yangjing1003@tjh.tjmu.edu.cn (J.Y.); yuzhang497@163.com (Y.Z.)

**Keywords:** mitochondrial respiratory chain, chronic renal failure, potential treatment strategies

## Abstract

The function of the respiratory chain is closely associated with kidney function, and the dysfunction of the respiratory chain is a primary pathophysiological change in chronic kidney failure. The incidence of chronic kidney failure caused by defects in respiratory-chain-related genes has frequently been overlooked. Correcting abnormal metabolic reprogramming, rescuing the “toxic respiratory chain”, and targeting the clearance of mitochondrial reactive oxygen species are potential therapies for treating chronic kidney failure. These treatments have shown promising results in slowing fibrosis and inflammation progression and improving kidney function in various animal models of chronic kidney failure and patients with chronic kidney disease (CKD). The mitochondrial respiratory chain is a key target worthy of attention in the treatment of chronic kidney failure. This review integrated research related to the mitochondrial respiratory chain and chronic kidney failure, primarily elucidating the pathological status of the mitochondrial respiratory chain in chronic kidney failure and potential therapeutic drugs. It provided new ideas for the treatment of kidney failure and promoted the development of drugs targeting the mitochondrial respiratory chain.

## 1. Introduction

The global total number of patients with acute kidney injury (AKI) and chronic kidney disease (CKD), and individuals receiving renal replacement therapy (RRT) has exceeded 850 million, posing a major burden on global public health [1,2]. Most kidney diseases, particularly chronic kidney disease, inevitably progress to end-stage renal failure. With a lack of effective drug treatment, life maintenance relies on dialysis or renal transplantation [3]. Patients have to face extremely high mortality and a heavy economic burden [4,5,6].

As the organ with the second highest oxygen consumption in the body at rest, the kidney has a high mitochondrial density second only to the heart [7,8].These physiological and structural characteristics are largely due to the tubular cells that comprise about 90% of the renal parenchyma, requiring a large supply of ATP to establish an energy-intensive electrochemical gradient [9,10]. Mitochondria rely on the respiratory chain to produce ATP [11], which closely links the normal functioning of the kidneys to the respiratory chain.

As the main source of ROS production, the mitochondrial respiratory chain plays an important role in the progression of kidney injury [12]. Simultaneously, the mitochondrial respiratory chain is also a major target of multiple uremic toxins (UTs) produced during kidney failure. Inhibition of the respiratory chain by UTs can lead to excessive ROS production, further promoting the production and accumulation of UTs, forming a positive feedback loop [13], and accelerating the deterioration of kidney function. Thus, the respiratory chain is closely linked with kidney failure.

This review aimed to summarize the most recent research progress on the relationship between the mitochondrial respiratory chain and chronic kidney failure, mainly elucidating the pathophysiological changes in the mitochondrial respiratory chain in chronic kidney failure and potential therapeutic drugs. It can provide new solutions for the treatment of chronic kidney failure and promote the development of drugs targeting the mitochondrial respiratory chain.

## 2. Mitochondrial Respiratory Chain in Normal Kidneys

### 2.1. Selection of Energy Substrates

Glucose and fatty acids are the two most commonly used energy substrates in normal kidneys. Fatty acid oxidation can produce three times more ATP than glucose oxidation, making it the preferred substrate for high-metabolic tissues/cells [14]. Different cells in a normal kidney exhibit varying mitochondrial densities and specific fuel preferences that often correspond to their ATP demands [15].

Glomerular cells, including podocytes, endothelial cells, and mesangial cells, mainly function to filter blood under the glomerular filtration pressure [16]. This passive process does not consume ATP, and ATP is primarily used to maintain cellular homeostasis. Therefore, glomerular cells mainly rely on glucose as an energy source and have a lower mitochondrial density [11,17,18]. In contrast, tubular cells primarily utilize fatty acids as their main energy source and have abundant mitochondria. This is because tubular cells (especially proximal tubule cells) actively reabsorb solutes and water from the primary filtrate, which is an energy-consuming process requiring significant ATP to maintain function [19,20].

### 2.2. Composition of the Mitochondrial Respiratory Chain

The mitochondrial respiratory chain, comprising five protein complexes located in the mitochondrial cristae and two electron carriers moving between these complexes, serves as the core of cellular energy metabolism [21]. The five protein complexes are identified as complex I (NADH-coenzyme Q reductase), complex II (succinate dehydrogenase), complex III (cytochrome bc1 complex), complex IV (cytochrome c oxidase), and complex V (ATP synthase), with the two electron carriers being coenzyme Q embedded in the membrane and soluble cytochrome c [21,22].

Complex I, the largest and most intricate component, facilitates the electron transfer between NADH and coenzyme Q, leading to the pumping of four protons across the membrane to provide the proton motive force essential for ATP synthesis [23,24,25]. Serving as a crossover point between the Krebs cycle and oxidative phosphorylation system, complex II catalyzes the oxidation of succinate to fumarate and transfers electrons to coenzyme Q. Despite not generating a proton motive force, it converts coenzyme Q to ubiquinol, thereby supplying fuel for complexes III and IV [26,27,28]. The ubiquinol generated by complexes I and II is further oxidized by complex III and the electrons are transferred to cytochrome c. This process is also coupled with the pumping of four protons into the intermembrane space [29]. Therefore, complex III is not only the balancing point for maintaining the redox state of the ubiquinone pool and the relay point for electron transfer between two electron carriers, but also the main source of proton motive force [30]. Complex IV accepts electrons transferred by cytochrome c to reduce oxygen to water. Additionally, while oxidizing one molecule of cytochrome c, it transfers two protons into the intermembrane space to accumulate proton motive force [31]. Finally, complex V utilizes the proton motive force accumulated by complexes I, III, and IV to phosphorylate ADP and generate ATP, which serves as the universal energy currency for cellular activities.

### 2.3. Generation and Function of Reactive Oxygen Species (ROS)

The production of ROS in kidney cells mainly occurs in the mitochondrial respiratory chain, generating highly active compounds such as hydrogen peroxide, superoxide anions, and hydroxyl radicals, collectively referred to as ROS [32,33]. Within cellular homeostasis, respiratory chain complexes I and III can generate low levels of ROS, primarily through the modification of key active Cys residues, participating in various intracellular signal transductions. These processes play a significant role in cell proliferation, differentiation, oxygen sensing, and mitochondrial autophagy [34,35,36].

## 3. Mitochondrial Respiratory Chain in Chronic Kidney Failure

### 3.1. Shift in Energy Substrates—Metabolic Reprogramming

In cases of renal failure, defects in the mitochondrial respiratory chain lead to disorders in ATP production, necessitating metabolic reprogramming of renal cells to adapt to injury and maintain ATP supply [37,38,39].

The most typical example is the proximal tubular cells of the kidney, which, as high-energy-demanding cells, are more susceptible to mitochondrial dysfunction, leading to an early shift from fatty acid oxidation to glycolysis to compensate for mitochondrial energy loss [40]. Additionally, when tubular cells are damaged, key upstream regulators of fatty acid oxidation such as peroxisome proliferator-activated receptor gamma coactivator 1 alpha (PGC1α), peroxisome proliferator-activated receptor alpha (PPARα), and the crucial enzyme for fatty acid oxidation, carnitine palmitoyltransferase 1A (Cpt1A), exhibit decreased expression. Consequently, insufficient fatty acid oxidation and an increase in free fatty acids occur [41,42,43]. The accumulation of free fatty acids, also known as non-esterified fatty acids (NEFA), within cells can have lipotoxic effects, mediating renal damage by promoting ROS production and activating NLRP3 inflammasomes and peroxisome proliferator-activated receptors (PPARs) [44,45,46,47]. Furthermore, in injured tubular cells, key rate-limiting enzymes of glycolysis and the major regulatory molecule, 6-phosphofructo-2-kinase/fructose-2,6-bisphosphatase 3 (PFKFB3), are activated. Sustained abnormal levels of glycolysis are strongly linked to renal fibrosis [48,49,50]. For instance, pyruvate kinase M2 (PKM2), one of the key rate-limiting enzymes of glycolysis, has been found to colocalize with the fibrosis marker vimentin in atrophic tubular cells. Moreover, the glycolytic metabolite lactate can maintain a positive feedback loop between glycolysis and fibrosis by sustaining the expression of the key rate-limiting enzymes hexokinase 2 (HK2) and PFKFB3 [51,52].

Consequently, while metabolic reprogramming initially enables the kidney to uphold energy production, long-term disruptions in lipid metabolism leading to lipid accumulation and lipotoxicity, as well as sustained incomplete energy compensation due to continuous glycolysis and fibrosis, are correlated with worsened renal outcomes.

### 3.2. Mitochondrial Respiratory Chain Dysfunction—“Poisoned” Respiratory Chain

The normal kidney plays a pivotal role in clearing metabolic waste generated within the body. However, in instances of renal failure, the inability to excrete various metabolic waste products leads to their accumulation in the body, disrupting normal physiological functions. These accumulated toxic compounds, referred to as UTs [53], are classified into three categories by the European Uremic Toxins Work Group (EUTox): water-soluble small molecules, medium-sized compounds, and protein-bound compounds [54].

Among the small molecule UTs, 4-hydroxynonenal (4-HNE), a lipid peroxidation product, has been extensively studied due to its ability to modify proteins through active double bonds and aldehyde groups, consequently impacting protein structure and function [55,56]. 4-HNE, generated and metabolized within the mitochondria, potentially targets the modification of respiratory chain protein complexes, inhibiting the activity of complexes I, II, and IV through precise modifications. This inhibition leads to reduced mitochondrial oxygen consumption and ATP production, increased proton leakage, and mitochondrial dysfunction [57,58,59,60,61,62,63,64]. Additionally, guanidinoacetic acid (GAA), another small-molecule UT, acts as a precursor molecule of creatine metabolism and can inhibit the activity of complexes II and IV, reducing ATP production, resulting in intracellular Ca^2+^ accumulation and cytochrome C detachment from the inner membrane, consequently leading to oxidative stress [65,66]. In the context of medium-sized compounds, resistin, a fat factor linked to obesity-mediated insulin resistance, exhibits a general inhibitory effect on the electron transport chain and complexes I, II, IV, and V, culminating in reduced ATP generation [67]. Similarly, tumor necrosis factor-α (TNF-α), another medium-sized UT closely associated with inflammation, invokes a decrease in the subunit expression of complexes I and III, demonstrating concentration-dependent inhibition of mitochondrial activity [68]. Furthermore, the protein-bound UTs, which form strong bonds with plasma macromolecular proteins making their removal challenging through dialysis, are notably linked to an increased incidence and mortality of patients with end-stage renal disease (ESRD) [69]. Among the prevalent protein-bound UTs, indoxyl sulfate, indoxyl-3-acetic acid, and p-cresol sulfate, metabolites of tryptophan, are highlighted, as even at low concentrations, they can hinder the activity of mitochondrial complexes III and IV enzymes, impeding energy generation in the respiratory chain [70,71]. Homocysteine (Hcy), a protein-bound UT derived from methionine metabolism, is considered an important independent risk factor for renal dysfunction [72]. Studies have shown a negative correlation between tissue Hcy levels and fifteen nuclear-encoded mitochondrial respiratory chain complex genes (encoding complexes I, IV, V), particularly indicating that high Hcy impedes complex assembly and reduces activity by inhibiting the core subunits S3/7 and V1/2 of complex I, ultimately leading to impaired electron transfer and an imbalanced mitochondrial redox state [73].

Mitochondria play a crucial role in the synthesis and metabolism of various pathways of UTs, and the mitochondrial respiratory chain complex stands out as a significant target for the damage caused by UTs. This interaction to a certain extent forms a positive feedback loop, accelerating the deterioration of kidney function [13].

### 3.3. Excessive Generation of ROS—Oxidative Stress

The imbalance between excessive generation of reactive oxygen species (ROS) and/or reduced antioxidant defense mechanisms is a major risk factor for the occurrence and development of kidney disease [74]. In renal failure, malfunction of the mitochondrial respiratory chain, alterations in the respiratory chain electron flux, and defects in downstream catalytic subunits result in electrons spending more time at the reduction center, leading to increased leakage and excessive ROS production [11,12,75]. The resultant excessive ROS are transferred to the mitochondrial matrix and intermembrane space from complexes I, II, and III, exacerbating the respiratory chain dysfunction, thus setting off a vicious cycle [76]. Notably, in patients with renal failure, plasma levels of non-enzymatic antioxidants such as vitamin C, vitamin E, selenium, and enzymatic antioxidants, including superoxide dismutase, catalase, glutathione, and glutathione peroxidase, are reduced. The imbalance between excessive oxidative reactions and an inadequate antioxidant system leads to renal oxidative stress (OS), causing cellular DNA damage, protein denaturation, and lipid peroxidation [77,78]. Of particular note is that although dialysis is currently the most effective treatment for ensuring the survival of chronic renal failure patients, apart from kidney transplantation, dialysis, especially hemodialysis, stimulates immune responses and inflammation by directly exposing blood to low biocompatible dialysis membranes and dialysate, exacerbating the production of ROS [79,80].

The pathophysiological changes in the mitochondrial respiratory chain in chronic renal failure are shown in Figure 1.

## 4. Kidney Failure Caused by Mitochondrial Respiratory Chain Abnormalities

Primary mitochondrial cytopathy encompasses a heterogeneous group of diseases resulting from mutations in nuclear DNA (nDNA) or mitochondrial DNA (mDNA), primarily leading to hereditary defects in the respiratory chain. Kidney disease occurs in approximately 5% of patients with primary mitochondrial cytopathy. However, due to the low prevalence of kidney biopsies in patients, this proportion is significantly underestimated [81,82].

Mitochondrial cytopathy can induce various renal phenotypes, with the renal tubule being the most commonly affected due to its high mitochondrial density. Proximal tubular involvement primarily manifests as Fanconi syndrome, which is the most frequent clinical presentation of mitochondrial cytopathy affecting the kidneys. This syndrome is characterized by reduced reabsorption of various filtered solutes [83]. Initial symptoms typically manifest in the neonatal period or before the age of 2, often accompanied by moderate renal failure and a poor prognosis for the patients [84,85,86,87]. Involvement of the distal renal tubules often results in electrolyte disturbances, particularly hypomagnesemia [88]. Mitochondrial cytopathy affecting the renal glomeruli commonly presents as nephrotic syndrome, especially steroid-resistant nephrotic syndrome, with focal segmental glomerulosclerosis being its typical histological feature [88,89,90]. Additionally, mitochondrial cytopathy may lead to tubulointerstitial nephritis and cystic kidney disease, ultimately culminating in end-stage renal disease [87,91,92].

Among the respiratory chain complexes, mutations associated with complex I deficiencies are the most prevalent primary respiratory chain disorders [93]. These mutations are categorized into two groups: one involves gene mutations encoding subunits of the complex, and the other encompasses gene mutations encoding auxiliary proteins vital for the biogenesis, assembly, and stability of the complex [94]. Phenotypes affecting the kidney as a result of alterations in complex subunits are often linked to the occurrence and metastasis of renal oncocytoma and renal clear cell carcinoma [95,96,97,98]. On the other hand, mutations in genes related to auxiliary proteins often lead to multi-system diseases, manifesting diverse kidney implications and early onset of kidney failure [92,94,99,100,101,102]. Additionally, the reduction in Ndufs6, as the second most conserved subunit in complex I, resulting in the lack of complex I, could also lead to isolated non-tumor-related kidney damage in young mice [103].

Complex II links the tricarboxylic acid cycle and serves as the entry point for succinate into the respiratory chain. Deficient activity of complex II leads to cellular energy characteristics resembling the Warburg effect, where cells depend on glycolysis to meet synthetic metabolic demands [104]. Mutations in complex II subunits are primarily associated with renal cell carcinoma of proximal tubule epithelial cell origin, representing the most prevalent kidney phenotype. Nonetheless, considering the fibrosis-promoting effect of glycolysis and the incomplete energy compensation, exploring kidney damage resulting from mutations linked to complex II is necessary [105]. In 2005, Goldenberg A and colleagues documented a case of congenital nephrotic syndrome in a patient with rapidly deteriorating renal function, culminating in end-stage renal failure by the age of 22 months. Despite the absence of related gene mutations, the patient’s clinical symptoms and the detection of complex II deficiency in multiple biopsied organs prompted the authors to contemplate primary complex II deficiency instead of secondary downregulation [106]. Additionally, Micheletti MV and colleagues documented a case where a mutation in the relevant gene led to recurrent hemolytic uremic syndrome and rhabdomyolysis due to complex II deficiency. This child required regular dialysis to sustain life from the age of two, largely supporting the evidence that complex-II-related mutations could lead to kidney failure [107].

The content and enzyme activity of mitochondrial complex IV are negatively correlated with the extent of tubular atrophy [108]. Hereditary defects in complex IV are one of the causes of chronic tubulointerstitial nephropathy [81,109,110]. Defects in complex IV are closely associated with the occurrence and development of chronic tubulointerstitial nephropathy. Renal Fanconi syndrome may be the initial sign of partial deficiencies in respiratory chain complex IV [110,111].

*ATP6* encodes a subunit of complex V, and its pathogenic mutations are currently the only ones known to be associated with kidney failure among the mutations in complex V. The pathogenic mechanism may be linked to mutations leading to ATP synthase dysfunction, indirectly affecting proton displacement in renal cells, and promoting ROS production [112,113]. However, as a target for renal ischemia–reperfusion injury and drug-induced kidney toxicity, there is still significant potential for exploring the connection between complex V and kidney failure [114,115].

Coenzyme Q plays a crucial role in maintaining the flow of reducing equivalents from complex I and II to complex III in the respiratory chain, and also serves as an essential antioxidant in the human body. In contrast to the tubular lesions commonly associated with the aforementioned respiratory chain complex deficiencies, hereditary defects in coenzyme Q often present as steroid-resistant nephrotic syndrome, with focal segmental glomerulosclerosis being the primary kidney pathology [116,117,118,119,120,121,122,123,124,125]. The specific mechanism of this unique glomerular lesion pattern remains unclear. However, in patients with *CoQ2* gene mutations, widespread abnormal mitochondrial proliferation was observed in glomerular cells. Furthermore, *Pdss2* knockout mice exhibited kidney diseases that were not observed in conditional *Pdss2* knockout mice in renal tubular epithelium, suggesting a certain degree of cell specificity in coenzyme Q deficiency-induced oxidative stress and mitochondrial dysfunction [119,122].

In renal failure, the abnormal internal environment of the body results in dysfunction of the mitochondrial respiratory chain. Defects in respiratory-chain-related genes are closely associated with the occurrence and development of kidney diseases. This provides a robust chain of evidence demonstrating the close relationship between the mitochondrial respiratory chain and the occurrence and development of kidney failure.

The characteristics of the renal failure caused by genetic defects in the respiratory chain are shown in Table 1.

## 5. Targeting the Mitochondrial Respiratory Chain: Potential Therapeutic Drugs for Chronic Renal Failure

### 5.1. “Starting from Scratch”—Targeting the Energy Substrate Selection Stage

In chronic renal failure, the primary pathophysiological mechanism at the energy substrate selection stage is the disorder in fatty acid utilization, and restoring normal fatty acid oxidation presents a potential therapy for this condition [19].

L-carnitine promotes the mitochondrial matrix transport of long-chain fatty acids, regulates fatty acid β-oxidation, and possesses antioxidant and free radical scavenging properties. It has exhibited benefits for various diseases characterized by low carnitine levels or impaired fatty acid oxidation [131,132,133]. In the human body, L-carnitine mainly relies on dietary intake and endogenous synthesis in the liver and kidneys. The balance of carnitine homeostasis is maintained through glomerular filtration and tubular reabsorption in the kidneys [134]. Chronic renal failure patients often suffer from carnitine deficiency due to dietary restrictions, renal dysfunction, and continuous loss during dialysis. Thus, oral or intravenous supplementation of L-carnitine presents a potential therapy for correcting the abnormal metabolic reprogramming in these patients [134,135]. Several clinical randomized controlled trials of L-carnitine supplementation have been conducted in patients undergoing hemodialysis or peritoneal dialysis for chronic renal failure; however, the existing results are somewhat controversial [136,137,138,139,140].

PGC-1α is a key transcription factor that regulates mitochondrial biogenesis and function, as well as a crucial upstream regulator of fatty acid oxidation [141,142]. Experimental models of chronic kidney disease in mice and patients with chronic renal failure exhibited low levels of PGC-1α expression. Pharmacological activation of PGC-1α presented a potential therapy for improving energy metabolism in patients with chronic renal failure [143,144,145,146,147]. A novel selective PGC-1α small-molecule agonist, ZLN005, has been validated in mice as promoting fatty acid oxidation and mitochondrial biogenesis. It was shown to improve insulin resistance and ketone tolerance in diabetic mouse models and to alleviate fibrosis and lipid accumulation in a unilateral ureteral obstruction (UUO) mouse model [148,149]. The traditional Chinese medicine prescription Shen Shuai II stimulated PGC-1α expression and improved mitochondrial functional protein expression and energy production in hypoxia-treated renal tubular epithelial cells (HK-2) and a 5/6 nephrectomy rat model of CKD. It also inhibited hypoxia-induced fibrosis in HK-2 cells [150]. Unfortunately, this treatment prescription lacks data related to fatty acid metabolism. The plant extract sulforaphane enhanced the expression of PGC-1α and nuclear respiratory factor 1 (NRF1) by suppressing the fatty acid intake membrane receptor CD36 and enhancing the expression of the key fatty acid oxidation enzyme CTP1A, reducing lipid deposition in a UUO rat model. It also improved the tricarboxylic acid cycle by increasing the expression and activity of mitochondrial functional proteins [151]. Furthermore, the plant extract huperzine A glucoside has been shown to activate PGC-1α transcription, but its specific pharmacological effects require further investigation [152].

The tissue expression of PPARα positively correlates with mitochondrial density and fatty acid β-oxidation levels, thus playing an important role in lipid metabolism [153]. Patients with chronic kidney failure exhibited reduced expression of renal PPARα. Mouse models further corroborated the link between low PPARα expression and the progression of fibrosis, suggesting that PPARα agonists hold potential as therapeutic drugs for chronic kidney failure [20,154,155]. Fibrates, the most commonly used PPARα agonists, are mainly excreted by the kidneys, thus limiting their use in patients with chronic kidney failure due to potential kidney-related complications. The novel fibrate pemafibrate, mainly excreted by bile, regulated fatty acid metabolism by activating renal PPARα and its target genes, leading to the inhibition of kidney fibrosis and the expression of inflammatory markers in UUO mice. Additionally, it improved plasma creatinine and blood urea nitrogen levels, as well as kidney fibrosis in CKD mouse models, consequently reducing renal inflammation and oxidative stress levels [156].

Cpt1A is a crucial rate-limiting enzyme in fatty acid metabolism. Reduced expression of Cpt1A in patients with chronic kidney failure is associated with fibrosis. Overexpression of Cpt1A in a mouse model of CKD restored fatty acid metabolism in the fibrotic kidney, which improved mitochondrial homeostasis and consequently ameliorated both renal fibrosis and kidney function [157]. Cpt1A agonists are potential drugs for targeting the energy substrate selection stage to improve fibrosis in chronic kidney failure. Resveratrol and its derivative, BEC2, have been experimentally confirmed as directly activating Cpt1A, thus accelerating long-chain fatty acid β-oxidation, but this class of drugs has not yet been used in animal CKD models [158,159].

### 5.2. Strive for “Precision Strike”—Targeting the Mitochondrial Respiratory Chain

Mitochondrial damage and dysfunction represent the primary pathogenic events in chronic kidney failure, with the dysfunction of the respiratory chain serving as the central component. Restoring the function of the mitochondrial respiratory chain is crucial in preventing the progression of chronic kidney failure [160].

Coenzyme Q10 (CoQ10) serves as both an electron carrier in the respiratory chain and an effective scavenger of reactive oxygen species [161]. The reduction in CoQ10 in the plasma of chronic kidney failure patients results in the diminished efficiency of electron transport in the respiratory chain, alterations in mitochondrial membrane potential, escalated production of reactive oxygen species, and a cascade of pathological changes [162]. As a result, the supplementation of CoQ10 not only enhances electron transport efficiency in the respiratory chain to facilitate ATP production, but also ameliorates abnormal fatty acid metabolism in diabetic and obese mice and patients with chronic kidney failure through the upregulation of PGC-1α expression. Additionally, it inhibits the depolarization of the mitochondrial membrane potential, thereby reducing oxidative stress markers in chronic kidney failure patients [162,163,164,165,166]. Moreover, CoQ10 supplementation demonstrated the ability to decrease proteinuria in a rat model of subtotal nephrectomy chronic kidney disease and in patients with primary CoQ10-induced kidney failure, consequently contributing to an improvement in kidney function to some extent. A large dosage of oral CoQ10 supplementation can successfully eliminate proteinuria and preserve normal kidney function in children with inherited mutations of *CoQ2*, *CoQ6*, and *CoQ8b* genes [90,123,167,168].

RP81-MNP is a nanocapsule-encapsulated renal enzyme stimulant that targets the proximal tubules of the kidney. RP81-MNP administration mainly upregulated the expression of mitochondrial respiratory chain complex I Nd1, Nd3–5 subunits and enhanced the reduction state of complex I to reduce cisplatin-induced renal tubular damage and excessive ROS production in a mouse model of CKD [169]. GC4419, a novel small molecule superoxide dismutase (SOD) mimic, demonstrated the ability to reduce excessive superoxide anion production induced by cisplatin. This was achieved by inhibiting the abnormal activity of mitochondrial respiratory chain complex I, leading to improvements in renal tubule necrosis, interstitial fibrosis, and the protection of kidney function in a mouse model of CKD [170].

Mitochondrial acid MA-5 is a newly synthesized indole derivative, which can regulate mitochondrial ATP synthesis and clear mitochondrial ROS production by promoting ATP synthase oligomerization and forming a supercomplex with mitofilin/Mic60 to improve mitochondrial dysfunction. Its nephroprotective effect has been further demonstrated in oxidative stress cell models and cisplatin-induced mouse nephropathy models [171,172,173]. The emergence of MA-5 provides a new strategy for mitochondrial-targeted therapy for chronic renal failure.

Mitochondrial complex I and cytochrome c are considered to be the targets of flavonoids [174]. Pre-administration of curcumin effectively mitigated the decline in respiratory chain complex I and V activities in a rat 5/6 nephrectomy model. This protective effect on the respiratory chain complex ameliorated excessive ROS production and renal structural damage. Unfortunately, the study on the efficacy of this drug has been limited to preventive administration [175,176,177]. Quercetin stimulated mitochondrial biogenesis and suppressed the production of reactive oxygen species by elevating the concentration of the electron carrier cytochrome c and inhibiting the generation of superoxide anions by mitochondrial complex I. Consequently, it suppressed inflammation and the expression of apoptosis factors in the rat UUO model [174,178,179]. Meanwhile, the mixed preparation of curcumin and quercetin, Oxy-Q, was confirmed in a phase I clinical trial to improve early graft function in deceased donor kidney transplant recipients. Further promotion of flavonoid preparations in the treatment of chronic kidney failure is anticipated [180].

The renal protective effect of non-flavonoid polyphenols, resveratrol, has been verified in various models of acute kidney injury [181,182]. Resveratrol could also improve mitochondrial ATP synthesis in the kidneys and reverse depolarization of mitochondrial membranes to alleviate glomerular injury in the 5/6 nephrectomy CKD rat model by increasing the expression of ATP synthase subunit beta and cytochrome c oxidase subunit I protein, and by exposing mesangial cells to TGF-β1 [183]. Unfortunately, the poor bioavailability of resveratrol has limited the translation of animal experiments to clinical trials. Improving the delivery of the drug, such as nanoencapsulation, is critical for its further clinical promotion [184].

Extracts of the traditional Chinese medicine formula Zhen Wu Decoction enhanced the expression of representative subunits NDUFB8, SDHB, UQCRC2, COX-I, and ATP5A of mitochondrial respiratory complexes I-V in the kidneys of mice in a UUO model, restoring oxidative phosphorylation and improving kidney fibrosis and renal function damage [185]. However, since this research was based on a composite formula, the specific effective ingredients need further clarification.

Preventive administration of the member of the vitamin E family, γ-tocotrienol, could effectively prevent the decrease in the activity of complexes I, III, and F_0_F_1_-ATPase after ischemia/reperfusion injury, preserve ATP levels in the renal cortex, and alleviate renal tubular injury and the post-injury inflammatory response [186]. However, as with curcumin, this drug is still in the stage of preventive administration and lacks verification in CKD models. It is unknown whether it has the same renal protective effect on patients with chronic kidney failure.

### 5.3. “Stepping on the Brake”—Targeting Mitochondrial Oxidative Stress

The causal relationship between oxidative stress and respiratory chain dysfunction is a primary contributor to the development and progression of chronic kidney failure. Addressing oxidative stress, particularly that originating from mitochondria, holds promise as a therapeutic approach for managing or ameliorating chronic kidney failure [74,76]. Traditional drugs that primarily act on energy substrate selection and mitochondrial respiratory chain stages can improve mitochondrial function to a greater or lesser extent while also having some degree of free radical scavenging effects. The primary hindrance to the antioxidant effect is the low concentrations of drugs in the mitochondria. The emergence of novel antioxidants specifically targeted at the mitochondria has facilitated the targeting of mitochondrial oxidative stress [12].

Mito molecules, such as MitoQ and Mito-TEMPO, are mitochondrial-targeted antioxidants traditionally linked to triphenylphosphonium (TPP) cations. Their specific delivery primarily relies on the electrostatic attraction between the outer TPP carrying a positive charge and the high transmembrane potential of the mitochondria [187]. MitoQ is formed by covalently connecting the quinone part to TPP. Upon entry into the mitochondria, the quinone part integrated into the mitochondrial lipid bilayer and underwent reduction by the respiratory chain, forming a quinol derivative. This derivative acted as a potent antioxidant, preventing lipid peroxidation and restoring activity through the respiratory chain cycle [188]. Currently, MitoQ has been validated to delay age-related kidney fibrosis in a mouse aging model and improve vascular dysfunction in patients with chronic kidney failure, suggesting its potential for application in chronic kidney failure patients [189,190]. Mito-TEMPO, an SOD mimic composed of peroxynitrite and TPP coupling, effectively reversed DNA methylation and reduced kidney fibrotic changes in an NDRG2-dependent manner, leading to a notable enhancement in renal function in a rat model of chronic kidney failure [191,192]. Furthermore, mitochondrial-targeted quinone analogs such as SkQ1 and SkQR1, as well as the SOD mimic Mito-CP, have demonstrated renal protective effects in various acute kidney injury models, though their verification in chronic kidney failure models is still pending [193,194].

Sodium tanshinone IIA sulfonate (SS) peptides are a class of cell-penetrating peptides with a specific mitochondrial-targeting sequence. They eliminate oxygen free radicals through tyrosine or dimethyltyrosine residues and are currently considered highly promising mitochondrial-targeted efficient antioxidants [195]. SS-31 penetrates the mitochondria in a manner independent of membrane potential and accumulates in the inner mitochondrial membrane to eliminate reactive oxygen species, thereby inhibiting the opening of mitochondrial permeability transition pores and the release of cytochrome c [196]. Administration of SS-31 effectively improved glomerulosclerosis and tubulointerstitial fibrosis in a rat 5/6 nephrectomy and unilateral ureteral obstruction (UUO) model, reduced renal function damage and proteinuria, and effectively prevented the transition from acute ischemic AKI to CKD [197,198,199,200]. SS-20, another SS peptide targeting the inner mitochondrial membrane, shares the same antioxidant mechanism as SS-31 but is not as widely utilized. While clearing mitochondrial reactive oxygen species, it effectively improved mitochondrial respiratory chain efficiency and ameliorated renal dysfunction and inflammation progression in a mouse model of chronic kidney failure [201]. Recently, electrostatically complexed SS-31 nanopolymer chains formed using anionic hyaluronic acid and cationic chitosan have achieved a breakthrough in targeting acute kidney injury after systemic administration, providing insights for targeting chronic kidney injury [202]. Additionally, mtCPP-1, a mitochondria-targeting peptide designed based on the structure of SS-31, has shown better mitochondrial-targeting ability than SS-31 [203]. Therefore, before clinical application in chronic kidney failure patients, the focus of drug improvement for SS-31 should be on improving the targeting of chronic kidney injury and mitochondrial targeting.

The potential therapeutic agents for chronic renal failure targeting the mitochondrial respiratory chain are shown in Table 2, and the mechanisms of action of the potential therapeutic drugs for chronic renal failure are shown in Figure 2.

## 6. Prospects

Although there are currently various drugs targeting different stages of the mitochondrial respiratory chain, most of them are limited in their further clinical application due to the difficulty in targeting the kidneys or mitochondria.

Currently, mitochondrial targeting primarily relies on TPP molecule linkage to enter the mitochondria in a membrane potential-dependent manner. Based on this, cobalt tetraoxide-polyethylene glycol-triphenylphosphine nanoparticles designed to target the mitochondria could induce mitochondrial autophagy and inhibit the transition from AKI to CKD [187,205]. The receptor for the giant protein on the surface of the proximal tubules in the kidney is a key target for kidney-specific delivery. Various biopolymer nanoparticles designed for this purpose have achieved targeted delivery to the renal tubules. For instance, the previously mentioned RP81-MNP and SS-31 electrostatically complexed nanopolymer chains are practical examples of nanoparticles encapsulated for proximal tubule targeting [169,202,206]. Moreover, recent research has focused on targeted drug delivery to the kidneys via extracellular vesicles. Vesicles derived from RAW264.7 could encapsulate dexamethasone and target the inflammatory renal tissue through their specific surface proteins integrin αLβ2 (LFA-1) and α4β1 (VAL-4) to mitigate the adverse effects of systemic dexamethasone administration [207].

In summary, developing drugs with triple targeting of the kidneys, mitochondria, and mitochondrial respiratory chain is the main direction for the research and development of a new generation of therapies for chronic kidney failure.

In addition, the emergence and more and more widespread use of multi-omics technologies (including genomics, proteomics, metabolomics, and transcriptomics) have provided powerful assistance in deepening the understanding of the mechanisms of chronic kidney disease progression and searching for pertinent biomarkers and potential therapeutic targets, enabling multi-dimensional and high-throughput evaluation of disease states and treatment effects [208]. In the future, multi-omics technology may empower precision therapy of chronic kidney failure with the combination of triple-target drugs (targeting the kidney, mitochondrial, and mitochondrial respiratory chain).

## 7. Search Strategy

We searched the PubMed, Embase, and Web of Science databases from inception to 13 August 2023. The search strategy combined the keywords and their variants of “mitochondrial respiratory chain” and “chronic kidney failure,” basic research. Clinical research, reviews, and case reports related to the mitochondrial respiratory chain and chronic kidney failure were included after reviewing their relevance. In addition, we manually searched the references of the identified studies in published reviews and cross-checked the search results as supplements.

## Figures and Tables

**Figure 1 ijms-25-00949-f001:**
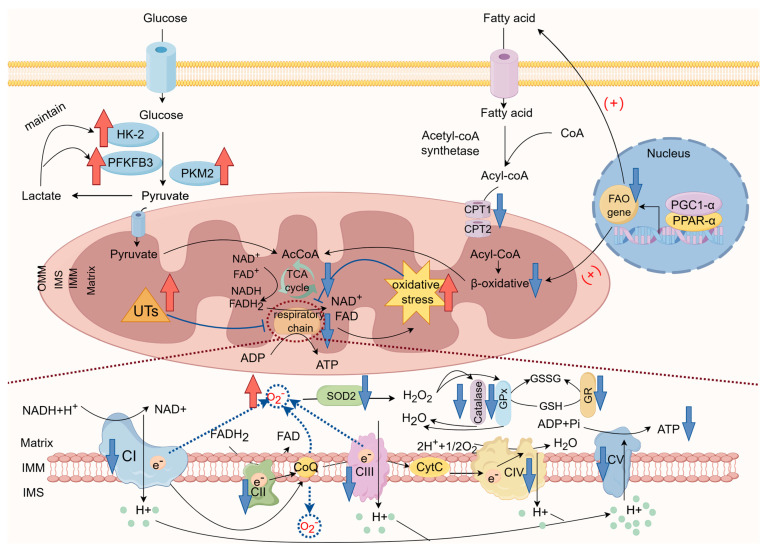
Mitochondrial respiratory chain in chronic kidney failure (by Figdraw). The mitochondrial respiratory chain complex is an important target for damage in chronic kidney failure. The production and accumulation of various uremic toxins (UTs) during kidney failure leads to dysfunction of the mitochondrial respiratory chain, resulting in inadequate energy generation. Kidney cells, especially proximal tubule cells, are forced to undergo metabolic reprogramming to adapt to the damage and maintain ATP supply. At this stage, the key upstream regulators of fatty acid oxidation levels, PGC1α, PPARα, and the key enzyme Cpt1A, show decreased expression, leading to lipid accumulation and kidney lipotoxicity. The expression of key rate-limiting enzymes in glycolysis, HK-2, PFKFB3, and PKM2, is enhanced, and sustained glycolysis results in incomplete energy compensation and accelerates kidney fibrosis. In addition, dysfunction of the respiratory chain leads to increased electron leakage at complexes I, II, and III, resulting in excessive ROS production. Meanwhile, levels of various antioxidant enzymes in the plasma of kidney failure patients are generally decreased, causing an imbalance between excessive oxidative reactions and inadequate antioxidant systems, leading to oxidative stress (OS) in the kidneys, exacerbating respiratory chain damage. AcCoA, acetyl-CoA; FAO, fatty acid oxidation; TCA, tricarboxylic acid cycle; IMM, inner mitochondrial membrane; IMS, intermembrane space; OMM, outer mitochondrial membrane; SOD2, superoxide dismutase2; GPx, peroxidase; GR, glutathione reductase; GSSG, oxidized glutathione; GSH, glutathione.

**Figure 2 ijms-25-00949-f002:**
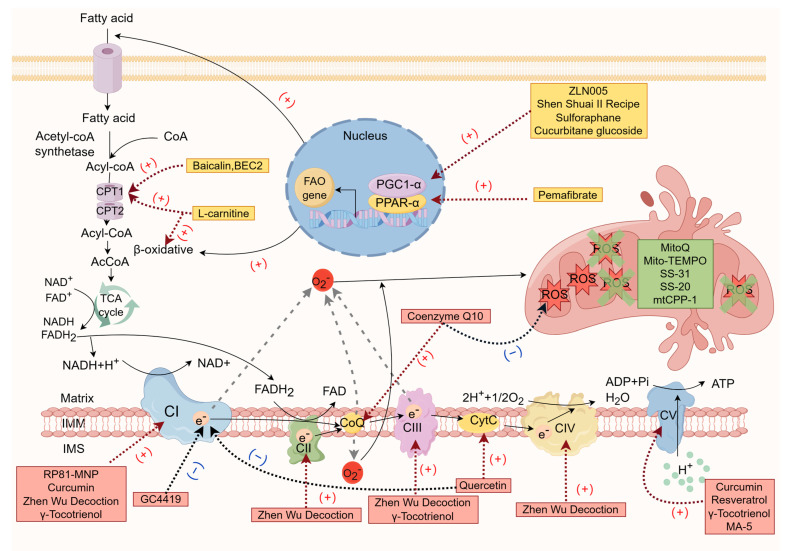
The mechanisms of action of potential therapeutic drugs for chronic renal failure (by Figdraw).

**Table 1 ijms-25-00949-t001:** Characteristics of the renal failure caused by genetic defects in the respiratory chain.

Defects of Respiratory Chain Components	Genetic Mutations	Species	Gene Function	Renal Phenotype and Time to Reach Renal Failure
Complex I	*TMEM126B* biallelic mutation, c.635G>T (p.Gly212Val) and/or c.401delA (p.Asn134Ilefs*2) [94]	Human	Encode component of the mitochondrial complex I assembly complex	Renal tubular acidosis in infancy, chronic renal insufficiency in early childhood
	*ACAD9* biallelic mutation, c.187G>T (p.E63*) and c.941T>C (p.L314P) [99,100]	Human	Encode a critical assembly factor for complex I biogenesis	Proximal tubular mitochondrial hyperplasia and renal failure in newborns
	*MTND5* frameshift mutation, m.12425delA (p.Asn30ThrfsX7) [92]	Human	Unclear as to the exact contribution to complex I function, but mutation leads to complex I assembly defects	Glomerular cystic disease with marked atrophy and fibrosis, school-age renal failure
	*XPNPEP3* homozygous frameshift, 931_934del AACA (p. N311LfsX5) [101,102]	Human	Maintain complex I stability	NPHP-like nephropathy, school-age renal failure
	*Ndufs6*^GT/GT^ (knockdown of the Ndufs6 gene) [103]	Mouse	Encode complex I subunits	Juvenile mice with increased urinary Kim-1 excretion and elevated circulating cystatin C
Complex II	Lack of coverage [106,107]	Human	The presence of relevant gene mutation and its function have not been cleared, but strongly associated primarily with the lack of complex II	Congenital nephrotic syndrome or hemolytic uremic syndrome, end-stage renal failure in early childhood
Complex III	*BCS1L* compound heterozygous mutation, c.166C>T (p.Arg56) andc.205C>T (p.Arg69Cys) [126]; homozygous mutation c.142A>G (p.M48V) [127]; homozygous mutation c.296C>T (P99L) [128,129,130];	Human	Encode chaperone/translocase that promotes Rieske Fe/S protein insertion in complex III	Fanconi syndrome; patients with compound heterozygous mutation had end-stage renal failure at age 49; patients with p.M48V homozygous mutation had renal failure at age 17; patients with homozygous P99L mutation had renal failure in newborns
Complex IV	*MTCO1* heteroplasmic nonsense mutation, m.6145G>A (p. Trp81Ter) [81]	Human	Encodes the core subunit of complex IV	Chronic tubulointerstitial disease, with abnormal mitochondria in distal tubular epithelial cells, chronic renal insufficiency in middle age
	A large-scale 7.3 kb deletion of mtDNA [110]	Human	Encodes three mitochondrial coding core subunits of the cytochrome c oxidase	Fanconi syndrome, chronic interstitial nephritis, preschool renal failure
	2.7 kb mtDNA deletion located between nucleotide (nt) 9700 and nt 13700 [109]	Human	Not directly involved in encoding cytochrome c oxidase, encoding tRNA affects the activity of cytochrome c oxidase	Fanconi syndrome, progressive renal insufficiency, tubular atrophy and interstitial fibrosis, extreme mitochondrial malformation of renal tubular cells, preschool renal failure
Complex V	*MT-ATP6* novel heteroplasmic truncating variant, m.8782 G>A (p. Gly86*) [112]	Human	Encodes ATP synthase subunit	Focal segmental glomerulosclerosis, renal failure in middle age
	*MT-ATP6* G8969>A mutation [113]	Human	Encodes ATP synthase subunit	Severe IgA nephropathy, multiple recurrences under steroid therapy, adolescent kidney failure
Coenzyme Q	*CoQ2* missense mutation, c.890G>A (p.Tyr297Cys) [116]; *CoQ2* frameshift mutations, c.1198delT, N401fsX415 [117]; heterozygous mutation, c.590G>A (p.Arg197His) and c.683A>G (p.Asn228Ser), homozygous mutation, c.437G>A (p.Ser146Asn) [119]	Human	Encodes key enzyme for coenzyme Q biosynthesis: para-hydroxybenzoate-polyprenyl transferase	Nephrotic syndrome (steroid resistance is common), focal and segmental glomerulosclerosis or crescent nephritis, renal failure in infancy and early childhood or adolescence
	*CoQ2* heterozygous mutation, c.1058A > G (p.Y353C) and c.973A > G, (p.T325A)	Human	Encodes key enzyme for coenzyme Q biosynthesis: para-hydroxybenzoate-polyprenyl transferase	Isolated nephrotic syndrome (steroid-resistant), preschool renal failure
	*CoQ6* homozygous mutation, c.763G>A (p.G255R), c.1058C>A (p.A353D) [120,121]	Human	Encodes coenzyme Q10 monooxygen6, catalyze cyclohydroxylation steps, which are required for CoQ biosynthesis	Steroid-resistant nephrotic syndrome, focal and segmental glomerulosclerosis, median age of renal failure less than 3 years
	*Pdss2*^kd/kd^ mice [122]	Mouse	Encodes decaprenyl diphosphate synthase subunit 2 for coenzyme Q biosynthesis	Nephrotic proteinuria, renal failure after 8 weeks
	*ADCK4* homozygous mutation, c.293T>G (p.Leu98Arg), c.1199dupA (p.His400Glnfs*11), c.1339dupG (p.Glu447Glyfs*10), c.1430G>A (p.Arg477Gln), c.293T>G (p.Lys98Arg) [123]; heterozygous mutation, the following genes are combined c.449G>A (p.R150Q), c.737G>A (p.S246N), c.532C>T (p.R178W), c.538C>T (p.R180C), c.551A>G (p.D184G), c.748G>C (p.D250H), c.936–938delGGT (p.V313del), c.1468C>T (p.R490C) [124]; compound heterozygous mutations, cc.439T>C (p.Cys147Arg) and c.1035 + 2T>C (p.?) [125]	Human	Gene involved in endogenous CoQ10 biosynthesis in human	Subnephrotic proteinuria or nephrotic syndrome (steroid resistance is common), renal failure usually occurs during adolescence or school age
Cytochrome C	NA	NA	NA	NA

**Table 2 ijms-25-00949-t002:** Potential therapeutic agents for chronic renal failure targeting the mitochondrial respiratory chain.

Drug Name	The Main Action Stage	Mechanism	Current Usage Status
L-carnitine [131,132,133,134,135,136,137,138,139,140,204]	Energy substrate selection	Mediates fatty acid transport and promotes the tricarboxylic acid cycle	Validated by randomized clinical trials in patients with hemodialysis and peritoneal dialysis with chronic renal failure, but the results were controversial
ZLN005 [148,149]	Energy substrate selection	PGC-1α agonists, promotes fatty acid oxidation, mitochondrial biogenesis and function	Phenotypic improvement validation of mouse model of diabetes mellitus and UUO
Shen Shuai Ⅱ recipe [150]	Energy substrate selection	Activates PGC-1α and regulates mitochondrial dynamics	Phenotypic improvement validation of rat 5/6 nephrectomy CKD model
Sulforaphane [151]	Energy substrate selection	Enhances PGC-1α and NRF1 expression, improves lipid metabolism and mitochondrial biogenesis	Phenotypic improvement validation of rat UUO model
Cucurbitane glucoside [152]	Energy substrate selection	Activates PGC-1α	Lack of animal model validation
Pemafibrate [156]	Energy substrate selection	PPARα agonist, regulates fatty acid metabolism	Phenotypic improvement validation of mouse UUO and purine-induced CKD models
Baicalin, BEC2 [158,159]	Energy substrate selection	CPT1A agonist, accelerates β oxidation of long-chain fatty acids	Not verified by mouse CKD model
Coenzyme Q10 [90,123,162,163,164,165,166,167,168]	Mitochondrial respiratory chain	improves the electron transport efficiency of the respiratory chain, activates PGC-1α to improve fatty acid metabolism, and inhibits mitochondrial membrane potential depolarization	Phenotypic improvement validation of a rat renal hemirectomy CKD model and patients with chronic renal failure, large-scale clinical randomized controlled trials were lacking
RP81-MNP [169]	Mitochondrial respiratory chain	Upregulates the expression of mitochondrial complex I subunit and enhances the reduction state of complex I	Phenotypic improvement validation of cisplatin-induced mouse CKD model
GC4419 [170]	Mitochondrial respiratory chain	Inhibits mitochondrial complex I aberrant activity	Phenotypic improvement validation of cisplatin-induced mouse CKD model
MA-5 [171,172,173]	Mitochondrial respiratory chain	Promotes ATP synthase oligomerization and forms a supercomplex with mitofilin/Mic60	Phenotypic improvement validation of cisplatin-induced mouse nephropathy model
Curcumin [176,177]	Mitochondrial respiratory chain	Maintains complexes I, V activity	Prophylactic administration was used to verify the protective effect of renal function in rat 5/6 nephrectomy CKD model
Quercetin [174,178,179]	Mitochondrial respiratory chain	Enhances cytochrome C concentration and inhibits the generation of superoxide anion by complex I	Validation of phenotypic improvement in rat UUO model
Resveratrol [183]	Mitochondrial respiratory chain	Increases the expression of ATP synthase β and cytochrome c oxidase subunit I protein, promotes ATP synthesis, and reverses mitochondrial hyperpolarization membrane potential	Validation of phenotypic improvement in rat 5/6 nephrectomy CKD model
ZhenWu Decoction [185]	Mitochondrial respiratory chain	Enhances mitochondrial respiratory complex I-V subunit expression to restore oxidative phosphorylation	Validation of phenotypic improvement in rat UUO model
γ-Tocotrienol [186]	Mitochondrial respiratory chain	Maintains complex I, III and F0F1-ATPase activity	Prophylactic administration has only been shown to be effective in a mouse model of ischemia–reperfusion acute kidney injury
MitoQ [189,190]	Mitochondrial oxidative stress	Targets mitochondria to prevent lipid peroxidation	Validation of phenotypic improvement in mouse aging model and chronic renal failure patients, large-scale clinical randomized controlled trials were lacking
Mito-TEMPO [191,192]	Mitochondrial oxidative stress	SOD enzyme mimics targeting ROS-mediated hypermethylation of the NDRG2 promoter	Validation of phenotypic improvement in mouse UUO model and rat 5/6 nephrectomy CKD model
SS-31 [197,198,199,200]	Mitochondrial oxidative stress	Targets the inner mitochondrial membrane to scavenge mitochondrial oxygen radicals by tyrosine or dimethyltyrosine residues	Validation of phenotypic improvement in rat 5/6 nephrectomy and UUO model
SS-20 [201]	Mitochondrial oxidative stress	Targets the inner mitochondrial membrane to scavenge mitochondrial oxygen radicals by tyrosine or dimethyltyrosine residues	Validation of phenotypic improvement in mouse 5/6 nephrectomy model
mtCPP-1 [203]	Mitochondrial oxidative stress	Targets mitochondria to scavenge mitochondrial oxygen radicals by dimethyltyrosine residues	Lack of animal model validation

## Data Availability

Data will be made available on request.

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
