# Peer review of "Focus on Mitochondrial Respiratory Chain: Potential Therapeutic Target for Chronic Renal Failure"

_ijms, 2024, doi:10.3390/ijms25020949_

Round 1

Reviewer 1 Report

Comments and Suggestions for Authors

The Review by Wang et al., on the pathological state of the mitochondrial respiratory chain in chronic kidney failure and potential therapeutic drugs is well written and organized. Figures and Tables are clear and easy to read. References are sufficient and up-to date.

However, I would suggest to the authors to introduce a dedicated paragraph focusing on kidney manifestations of genetic mitochondrial-related cytopathies.

Reviewer 2 Report

Comments and Suggestions for Authors

The article by Yi Wang, Jing Yang, Yu Zhang and Jianhua Zhou, presents a review of the mitochondrial respiratory chain as a potential therapeutic target for renal failure.

The authors describe the mitochondrial electron respiratory chain (MRC) and its role in kidney failure (KF), along with how abnormalities in the MRC are involved in KF. They then go on to discuss some potential therapeutic approaches to overcome MRC dysfunction, in particular with respect to chronic renal failure.

The authors need to consider the following:

1)            The authors do not discuss mitochonic acid (MA-5) in their manuscript. I feel that this molecule is pertinent to the topic and should be mentioned.

2) The manuscript needs to be read by a native English speaker, in particular to correct the improper use of the verb tense in the manuscript.

Comments on the Quality of English Language

The manuscript needs to be read by a native English speaker, in particular to correct the improper use of the verb tense in the manuscript. In general:

-where something is known to be true/accepted, then the present simple tense should be used.

-the present perfect tense is used when describing something has been done in the past but may still be relevant in the present time.

-the past simple tense is used to describe that has been accomplished in the past

Reviewer 3 Report

Comments and Suggestions for Authors

General Comments.

This review assesses research related to the mitochondrial respiratory chain and chronic kidney failure, primarily elucidating the pathological status of the mitochondrial respiratory chain in chronic kidney failure and potential therapeutic drugs. Additionally, it provides new ideas for the treatment of kidney failure and promotes the development of drugs targeting the mitochondrial respiratory chain.

Specific Revision Comments:

1.     In the Introduction, the aim of the paper should be reported in this section.

2.     Please add a search strategy section, reporting how authors selected articles, time period, and keywords.

3.     In the Prospects section, innovative prospective "OMIC" techniques, namely genomics, proteomics, metabolomics, and transcriptomics, are excellent candidates to provide a better understanding of the mitochondrial respiratory chain and the associated risk of cardiovascular events.(Reference: Int.J.Mol.Sci.2022,23,336. https:// doi.org/10.3390/ijms23010336) Please discuss this point, adding the reference.

Comments on the Quality of English Language

Minor editing of English language required

Round 2

Reviewer 2 Report

Comments and Suggestions for Authors

The authors have addressed the points raised.